# The Solution Combustion Synthesis of ZnO Powder for the Photodegradation of Phenol

**Aleksandr P. Amosov** [1,*], **Vladislav A. Novikov** [1], **Egor M. Kachkin** [1], **Nikita A. Kryukov** [1], **Alexander A. Titov** [1], **Ilya M. Sosnin** [2] and **Dmitry L. Merson** [2]

1   Department of Physical Metallurgy, Powder Metallurgy, Nanomaterials, Faculty of Mechanical Engineering, Metallurgy and Transport, Samara State Technical University, Molodogvardeiskaya 244, 443100 Samara, Russia

2   Research Institute of Advanced Technologies, Togliatti State University, Belorusskaia 14, 445020 Togliatti, Russia

*   Correspondence: egundor@yandex.ru

**Abstract:** Nanoscale and submicron powder of zinc oxide (ZnO) is known as a highly efficient photocatalyst that is promising for solving the problem of wastewater treatment from toxic organic pollutants including phenol and its derivatives. The results of laboratory studies of ZnO preparation by a simple, energy-saving, and highly productive method of solution–combustion synthesis from a mixture of solutions of zinc nitrate and glycine, as well as the use of the ZnO powder synthesized by combustion for the photocatalytic decomposition of phenol, are presented. The modes and characteristics of combustion, phase composition, chemical composition, and structure of the combustion product at different ratios of glycine with zinc nitrate were determined. It is shown that calcination at 650 °C reduces the content of carbon impurity in the combustion product to ~1 wt.% and leads to obtaining ZnO powder in the form of porous agglomerates up to 100 μm in size sintered from crystalline nanoscale and submicron ZnO particles with an average crystallite size of 44 nm. The ZnO powder exhibits high photocatalytic activity, leading to the almost complete degradation of phenol in an aqueous solution under the action of ultraviolet irradiation in less than 4 h.

**Keywords:** zinc oxide; powder; solution combustion synthesis; photocatalysis; phenol; degradation

## 1. Introduction

Zinc oxide (ZnO), with its unique physical and chemical properties, is an important functional ceramic material [1]. ZnO is a semiconductor with a wide band gap energy (3.37 eV); high bond energy (60 meV); photoluminescent, piezo-, and pyroelectric properties; and high thermal and mechanical stability, which make it attractive for potential use in electronics, optoelectronics, photonics, acoustics, sensing, solar cells, and laser technology [1–3]. Ultrafine, i.e., nanoscale and submicron, zinc oxide is also known as a highly efficient catalyst for various organic reactions, as well as a photocatalyst for the oxidation of toxic organic substances consisting of elements such as C, H, and O to harmless $CO_2$ and $H_2O$ [4,5]. The latest application of zinc oxide attracts the increased attention of researchers to solving the problem of wastewater treatment from toxic organic pollutants including phenol and its derivatives that are resistant to environmental degradation and have a harmful effect on humans, animals, and the environment [5,6]. Phenol $C_6H_5OH$ is one of the most toxic and highly dangerous substances. Such traditional water treatment techniques as adsorption, membrane separation, and coagulation only concentrate or change the organic pollutants from the water to the solid phase. Because of this, additional cost and treatments are needed to treat the secondary pollutants and regenerate the adsorbents. Semiconductor photocatalysis is one of the most important and promising approaches due to its high efficiency, simplicity, nontoxicity, low energy consumption, and economy [7,8].

The photocatalytic effect is explained by the fact that in a semiconductor, electrons are in free (conduction band) and bound (valence band) states. The transition of electrons from the valence band to the conduction band is associated with energy consumption governed by the band gap energy (3.37 eV for ZnO), which can be provided by light quanta with the appropriate energy. As a result of light absorption, free electrons and holes are formed that, moving in the crystal lattice, partially recombine and partially come to the surface. Being extremely reactive, electrons and holes interact with water and oxygen on the surface of semiconductor particles, which leads to the formation of hydroxyl radicals and a series of highly reactive oxygen-containing compounds that interact with phenol $C_6H_5OH$, decomposing it and converting it into safe $CO_2$ and $H_2O$ [5–8]. The activity of a photocatalyst is dictated by its ability to create photogenerated electron–hole pairs.

Among semiconductors, titanium dioxide ($TiO_2$) has been the most studied compound in processes of photodegradation of organic compounds. $TiO_2$ possesses a large band gap energy of 3.2 eV, and thus, high-energy ultraviolet (UV) irradiation is needed to activate it, making it difficult to use for large-scale, economically viable photocatalytic applications [5]. Research is now focusing on using solar energy for the photocatalytic degradation of organic pollutants in water. The amount of UV irradiation in solar light is about 5% of the spectrum, while 45% of sunlight is in the visible light region [9]. The application of $TiO_2$ using solar energy is highly restricted by its large band gap and low quantum efficiency [10]. ZnO has been proposed as an alternative photocatalyst to $TiO_2$ as it possesses the same band gap energy but exhibits higher absorption efficiency across a large fraction of the solar spectrum and 75% lower production cost when compared with $TiO_2$ [5,11,12]. (For the sake of fairness, however, it should be noted that research on the improvement of $TiO_2$ photocatalysts continues. For example, a flower-like $TiO_2$-based composite (denoted as Zn-Ti-6) showed better adsorption and performance of methylene blue photocatalytic degradation than $TiO_2$ nanoparticles owing to its much larger specific surface area, more abundant hydroxyls, and lower photoluminescence intensity [13].)

The major constraint of ZnO is the rapid recombination rate of photogenerated electron–hole pairs, which perturbs the photodegradation reaction. The solar energy conversion performance of ZnO is limited by its large band gap energy. To improve the performance of ZnO in photocatalysis, intense efforts have been made to minimize the band gap and inhibit the recombination of photogenerated electron–hole pairs [5–7,14]. Various methods of modifying ZnO have been developed: metal/non-metal doping, coupling with other semiconductors and elements, crystal growth and shape control, and surface defects control [5,7,14–19]. In high demand are ultrafine forms of zinc oxide including nanoparticles, rods, and films. Considering the need for ZnO powders of various morphology, the search for different production methods and the study of conditions and formation mechanisms in ultrafine state remain an urgent challenge [20]. Such methods of synthesis and modification will enhance the ZnO performance in photocatalysis by shifting the band gap energy, suppressing the recombination rate of electron–hole pairs, increasing charge separation efficiency, improving production rate of hydroxyl radicals, producing smaller particle size with high specific surface area, and allowing better dispersion in medium [5,14].

A variety of vapor phase and solution-based methods for ZnO production in nanoscale and submicron states have been developed [5]. Gas-phase methods are represented by thermal evaporation, pulsed laser deposition, chemical and physical vapor deposition in various variants, and molecular beam epitaxy. They are complex, implemented on expensive equipment, energy-intensive, and inefficient. Solution methods include hydrothermal, solvothermal, sol-gel, precipitation, microemulsion, microwave, electrochemical, and others. Solution methods are simpler and less energy-intensive, but they allow for effectively regulating the composition, morphology, and size of synthesized ZnO nanopowders by such factors as the type of solvent, composition of reagents, and synthesis conditions. Many of these methods allow forsynthesizing ZnO nanostructures of various types, but all of

them are characterized by low productivity, which prevents the industrial production of ZnO-nanostructured photocatalysts for wastewater treatment.

The solution–combustion synthesis (SCS) of oxides, which appeared relatively recently, differs markedly from the methods listed above in its simplicity, energy saving, and high productivity, which makes it attractive for the creation of industrial production technologies for relatively inexpensive oxide nanomaterials of various applications [20–23]. In order to achieve practical application, ZnO nanopowder should be cheaper than other photocatalysts; therefore, it is of interest to study the possibility of obtaining ZnO by the SCS method and determining its photocatalytic activity. Some results of studies on the possibility of obtaining ZnO by the SCS method are known, but they do not describe the combustion process and products in sufficient detail, which makes it difficult to choose them for reasonable practical application for the photocatalytic degradation of phenol [20,24–28]. The SCS process is based on the combustion of mixtures of dissolved reagents of highly exothermic redox reactions. The purpose of this work is to study in detail the process and products of combustion during the synthesis of a ZnO nanoscale and submicron powder from a mixture of solutions of such common reagents as the oxidizer zinc nitrate ($Zn(NO_3)_2$) and the reducing agent (fuel) glycine ($C_2H_5NO_2$), as well as for the first time studying the use of the ZnO powder synthesized by combustion for the photocatalytic decomposition of phenol.

## 2. Materials and Methods

During experimental studies, such materials as zinc nitrate hexahydrate ($Zn(NO_3)_2 \cdot 6H_2O$, $\geq 98$ wt.%, Ural Plant of Chemical Products, Russia), glycine ($C_2H_5NO_2$, $\geq 99.5$ wt.%, Khimreaktiv, Russia), and phenol ($C_6H_5OH$, $\geq 99.50$ wt.%, Kazanorgsintez, Russia) were used as reagents.

The equation of the ZnO synthesis reaction using the selected reagents is set up from the calculation of the oxidative and reducing valences of the reagents and has the following form [23,26,29]:

$$Zn(NO_3)_2 + \frac{10}{9}\varphi\, C_2H_5NO_2 + \frac{5}{2}(\varphi - 1)O_2 = ZnO + \frac{25}{9}\varphi H_2O + \frac{20}{9}\varphi CO_2 + (\frac{5}{9}\varphi + 1)N_2 \qquad (1)$$

where the dimensionless parameter $\varphi$ is equal to the molar ratio of fuel to oxidizer. The $\varphi$ parameter characterizes the redox mixture of reagents as stoichiometric at $\varphi = 1$, when the internal atomic oxygen in the redox mixture is sufficient for complete oxidation of the fuel, and no external (atmospheric) molecular oxygen $O_2$ is required for this. When $\varphi < 1$ (fuel-lean mixtures), the internal atomic oxygen in the redox mixture is more than enough for complete oxidation of the fuel, and molecular oxygen is even produced during combustion. At $\varphi > 1$ (fuel-rich mixtures), the internal atomic oxygen in the redox mixture is not enough for complete oxidation of the fuel, and the external molecular oxygen must be consumed during the combustion of the redox mixture. (Note that anhydrous zinc nitrate ($Zn(NO_3)_2$) appears in Equation (1), since when the solution of the reagent mixture is heated, almost all free and bound water evaporates, and the synthesis reaction takes place during combustion of an almost anhydrous viscous gel.) The value of the $\varphi$ parameter largely governs the combustion mode, as well as the composition and structure of the combustion products. The experimental study of the combustion process and products was carried out when the values of the $\varphi$ parameter changed in the range $0.25 \leq \varphi \leq 3$ in increments of 0.25.

Required amounts of zinc nitrate ang glycine were dissolved in distilled water until saturated and mixed at given fuel-to-oxidant ratios $\varphi$. The reagent mixture solution with a volume of 25 mL and a flat layer height of 6.5 mm was heated in a metal reaction vessel with a flat bottom on an electrical hot plate with a power of 1 kW and a surface temperature of 460 °C. Heating of the solution led to a relatively slow increase in temperature from the initial value to the boiling point and subsequent boiling away of the main amount of free and bound water with the resulting formation of a viscous mixture of reagents (gel). Then

the temperature of the gel increased rapidly due to the spontaneous onset of a chemical reaction with intense heat and gas release culminating in combustion. Upon completion of the combustion, a loose or dense cake of solid combustion product remained in the reaction vessel, and the mass of the product depended on the mode of combustion. Intensive combustion led to the ejection of a part of the reacting mixture and combustion products from the vessel, so that only a part of the solid combustion product remained in the vessel. In this regard, the coefficient of product mass conservation $K_m$ was calculated as the ratio of the mass of combustion product remaining in the reaction vessel after the experiment to the theoretical mass of the solid product calculated by the reaction equation. Along with the determination of the mode of combustion and the calculation of the $K_m$ coefficient, the temporal characteristics were determined: (1) the delay time of the beginning of combustion (ignition) $t_{ign}$ from the start of heating and (2) the duration of combustion $t_{comb}$. (Note that such characteristics of the SCS process as $K_m$, $t_{ign}$, and $t_{comb}$ are being introduced into consideration for the first time.)

The phase composition of the synthesized products was analyzed using an ARL X'TRA X-ray diffractometer (Thermo Fisher Scientific, Basel, Switzerland) equipped with an X-ray tube with a copper anode with a maximum power of 2200 W. Scanning was performed in the $2\theta$ range of angles (20–80°) at a speed of 2 deg/min. The microstructure and elemental chemical composition were studied with a scanning electron microscope (SEM) JSM-6390A (Jeol, Tokyo, Japan) equipped with an energy-dispersive spectroscopy (EDS) module JSM-2200. Crystallite sizes were calculated from X-ray peak broadening using the Scherrer formula. The particle-size distribution of the synthesized powder was determined by laser diffraction of aqueous suspensions containing the particles using a Sald-2300 analyzer (Shimadzu, Kyoto, Japan). The band gap of the ZnO was measured by registering the edge of absorption of electromagnetic radiation using a UV-2600 UV-vis spectrophotometer (Shimadzu) equipped with an ISP-2600Plus integrating sphere (Shimadzu). The interpretation of the data obtained was carried out using the construction and analysis of the Tauc plots with the programs SPCTrans and Band Gap Calculation Excel Macro.

The study of the photocatalytic activity of synthesized ZnO was carried out in the decomposition of phenol dissolved in 100 mL of water at a concentration of 1 mg/L. ZnO particles were dispersed in a solution in an amount of 1 g/L using an ultrasonic bath UZV-2.8 (Sapphire, Moscow, Russia). The process of photocatalytic decomposition proceeded with constant stirring of the solution under the action of ultraviolet irradiation with a wavelength of 365 nm on the Lab 365 nm BLB TL-D 18W installation (Philips, Amsterdam, The Netherlands). The concentration of phenol dissolved in water was determined by registering a characteristic fluorescent peak with the RF-6000 fluorescence spectrometer (Shimadzu).

The presence of by-products of phenol oxidation in water after the photocatalytic reaction was investigated using a PE-5400 UV spectrophotometer (Promecolab, Saint-Petersburg, Russia). The spectral range of the device was 190 to 1000 nm, and the spectral resolution of the device was 1 nm.

## 3. Results and Discussion

### 3.1. Modes and Characteristics of Combustion

The results of experiments on the determination of the combustion characteristics for the mixture solution of fuel (glycine) with oxidizer (zinc nitrate) for different molar ratios of fuel to oxidizer ($\varphi$) are presented in Figure 1 in the form of graphical dependencies $t_{ign}(\varphi)$, $t_{comb}(\varphi)$, and $K_m(\varphi)$.

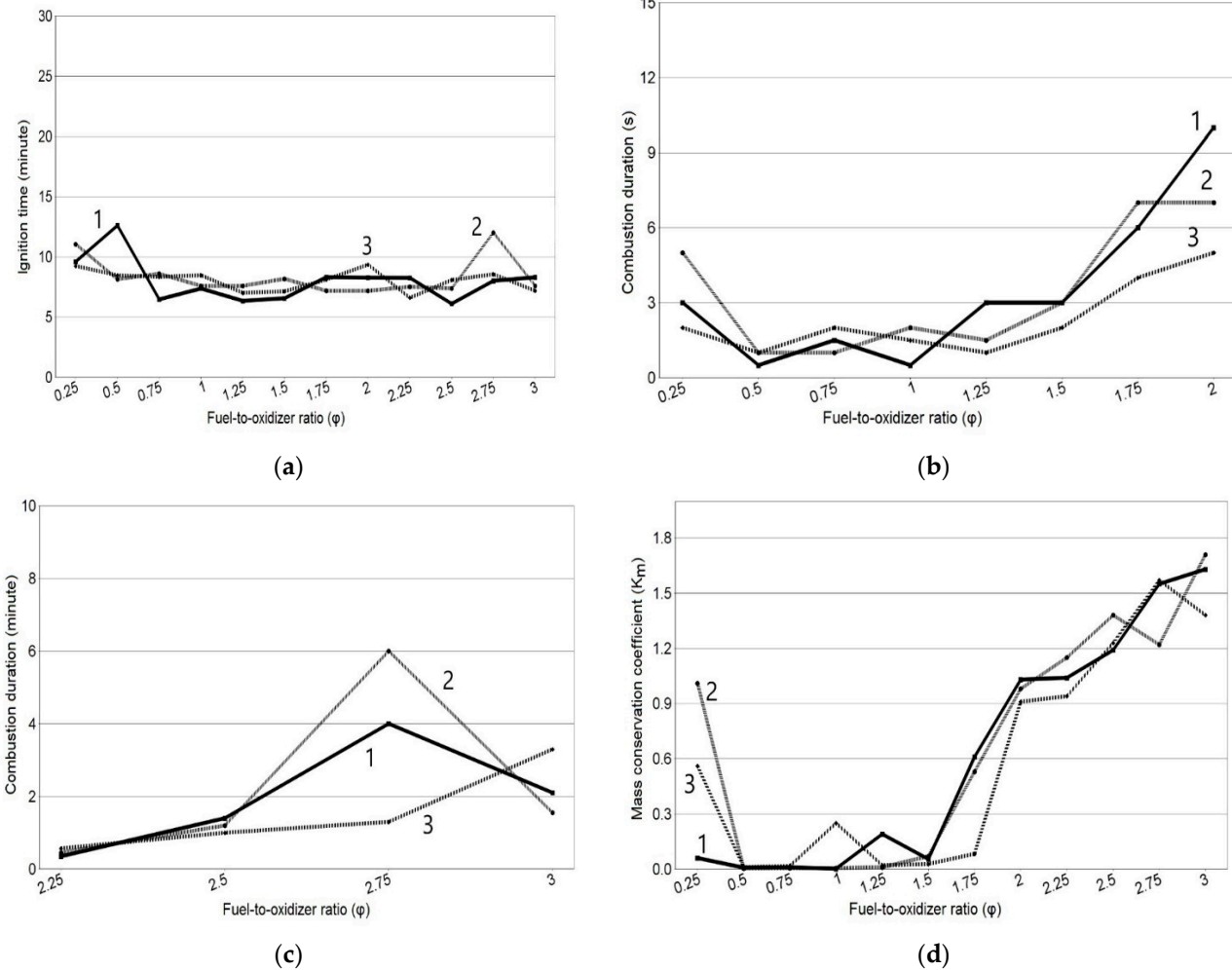

**Figure 1.** Dependence of the combustion characteristics for the SCS process on the $\varphi$ parameter: (**a**) $t_{ign}(\varphi)$; (**b**) $t_{comb}(\varphi)$ from $\varphi = 0.25$ to $\varphi = 2$; (**c**) $t_{comb}(\varphi)$ from $\varphi = 2.25$ to $\varphi = 3$; (**d**) $K_m(\varphi)$.

These results were obtained by conducting three experiments for each value of the $\varphi$ parameter. The numbers on the lines indicate the numbers of experiments, and it is clear that the combustion characteristics obtained in three experiments for the same value of the $\varphi$ parameter can differ significantly from each other; that is, they have a wide scatter. This scatter is due to the stochastic nature of spontaneous ignition on the relatively slow heating of the solution and then the gel of the reagent mixture.

According to the results obtained, the ignition delay time $t_{ign}$ from the start of heating to the start of combustion reaction weakly depends on the $\varphi$ parameter, averaging 8 min (Figure 1a). This is explained by the fact that it consists of the time of heating the solution to boiling water, the time of its almost complete evaporation, and the formation of a gel capable of self-ignition. For all values of the $\varphi$ parameter, these components of the time $t_{ign}$ are practically the same, but other characteristics and the mode of combustion change significantly when the $\varphi$ parameter changes (Figure 1b–d). In the $\varphi$ range of 0.5 to 1.5, the reaction takes place in the mode of extremely fast ($\leq 3$ s) volumetric combustion with the formation of a yellow flame; a sharp, almost complete ejection ($K_m \ll 1$) of the reacting mixture; and reaction products of milky color in the form of white dense smoke from the vessel. With reduced fuel content ($\varphi = 0.25$), the reaction takes place in a flameless mode without the formation of luminous zones but with a rapid release of red smoke for 2–5 s and a partial ejection ($K_m \leq 1$) of a light green product. At $\varphi > 1.5$, the mode of combustion and the product change again. There is a transition to synthesis in the mode of increasingly slower smoldering with the formation of spot and frontal luminous zones (on average

up to 3 min) with increasing φ. In this connection, the representation of the dependence $t_{comb}(\varphi)$ had to be divided into two figures: Figure 1b shows the time of rapid combustion expressed in seconds for φ from 0.25 to 2, and Figure 1b shows the time of slow combustion expressed in minutes for φ from 2.25 to 3. The color of the loose product changes from gray with an admixture of white to black with an admixture of white, and entirely black at φ = 2.5 or more, and the coefficient of conservation of product mass $K_m$ at φ > 2 even becomes greater than 1.

Such changes in the color and mass of the product are explained by the fact that at φ > 1.5, the mixture of reagents becomes fuel-rich, and therefore, the internal oxygen in the mixture of reagents and external oxygen from the ambient air is deficient for the complete oxidation of carbonaceous fragments of organic fuel and their removal from the combustion product in the form of gases [20,27]. If pure ZnO has a white color, then unburned glycine residues in the form of free carbon and carbon bound to other elements, mainly oxygen and hydrogen, have a black color [20,27]. With an increase in the φ parameter, the combustion product along with white ZnO contains more unburned glycine residues, acquires a gray color, and at φ > 2 becomes black in color; its mass becomes greater than the theoretical yield of ZnO, and therefore, the coefficient $K_m$ becomes greater than 1. In this case, in order to complete the reaction over the entire volume of the reagent mixture, this mixture has to be constantly stirred during smoldering. From these results, it can be concluded that the value of φ = 2 is most suitable for practical application, in which there is no explosive combustion with the ejection of the product from the vessel observed at lower φ, and intensive smoldering in about 8 s leads to the formation of an easily destructible powder with its preservation in the reaction vessel without the need for constant stirring required at φ > 2 to complete the synthesis reaction.

### 3.2. Composition and Structure of Combustion Products

The results of determining the phase and chemical composition as well as the microstructure of the SCS products for different values of the molar ratio of fuel to oxidizer (φ) are shown in Figures 2–4.

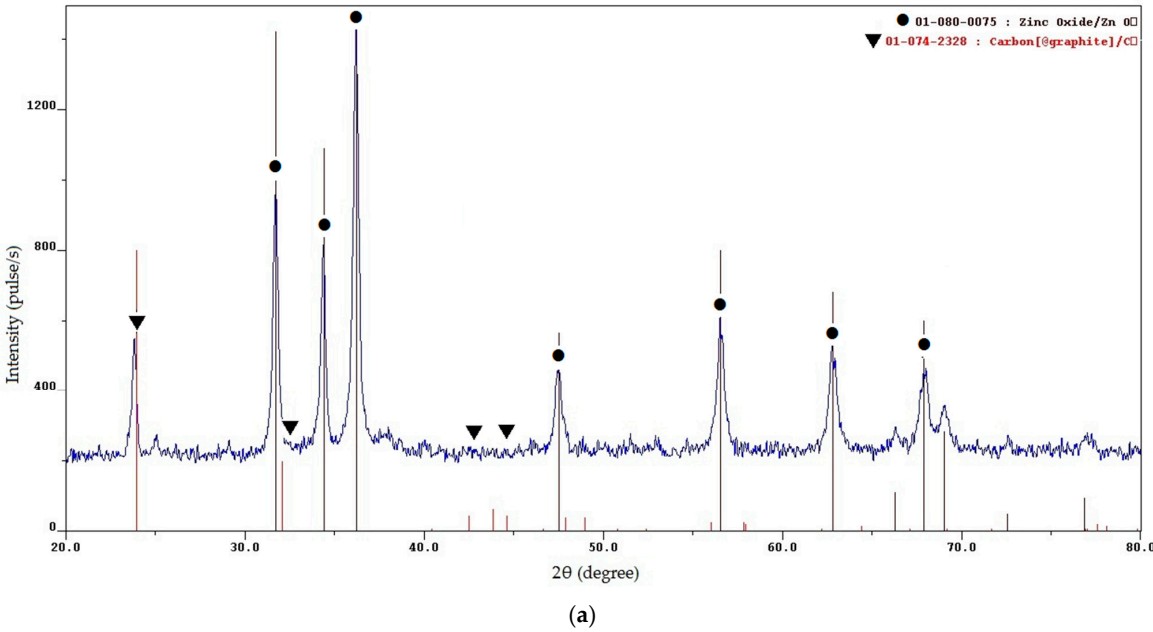

(a)

**Figure 2.** Cont.

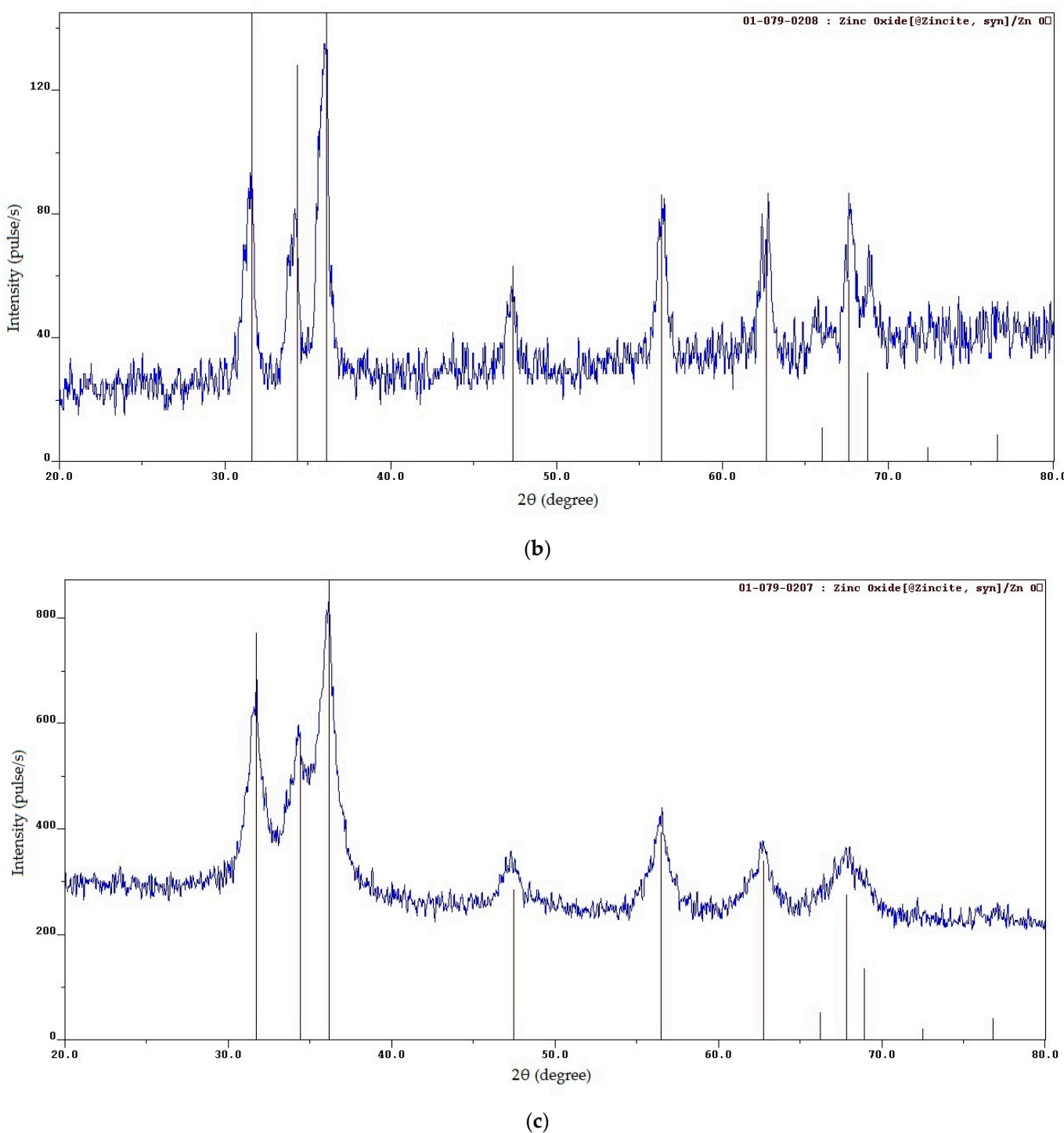

**Figure 2.** Cont.

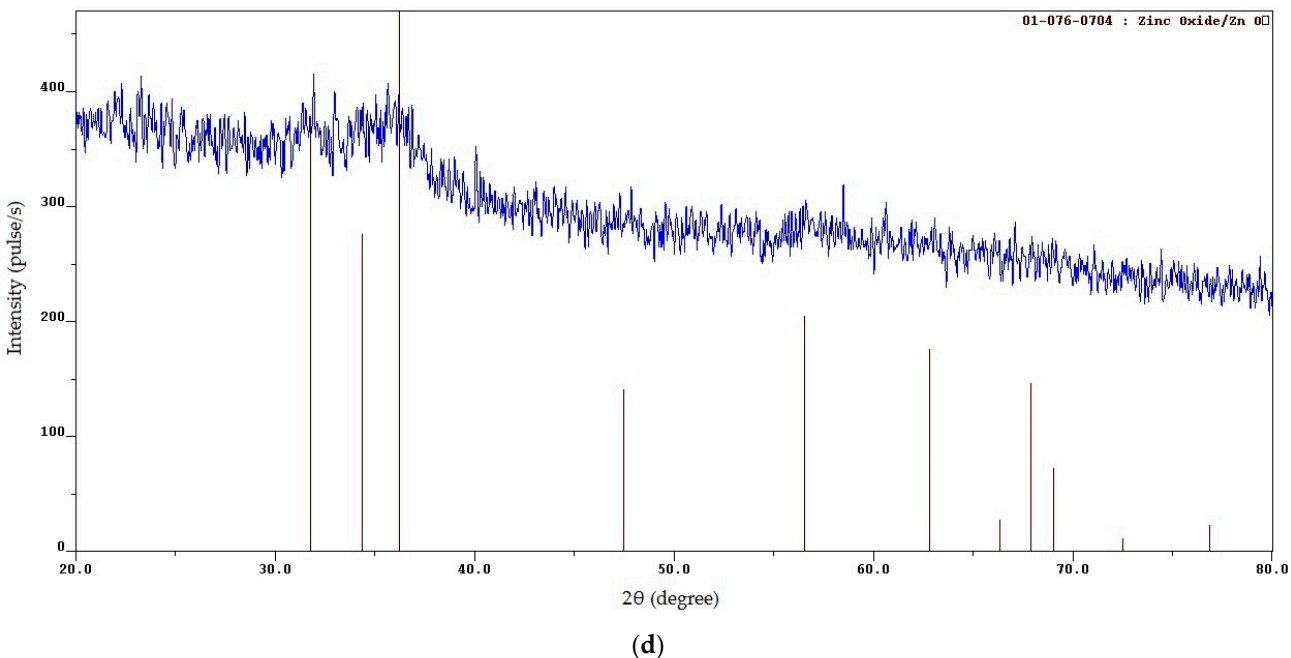

(**d**)

**Figure 2.** XRD patterns of the SCS product at various φ: (**a**) φ = 0.25; (**b**) φ = 1; (**c**) φ = 2; (**d**) φ = 3.

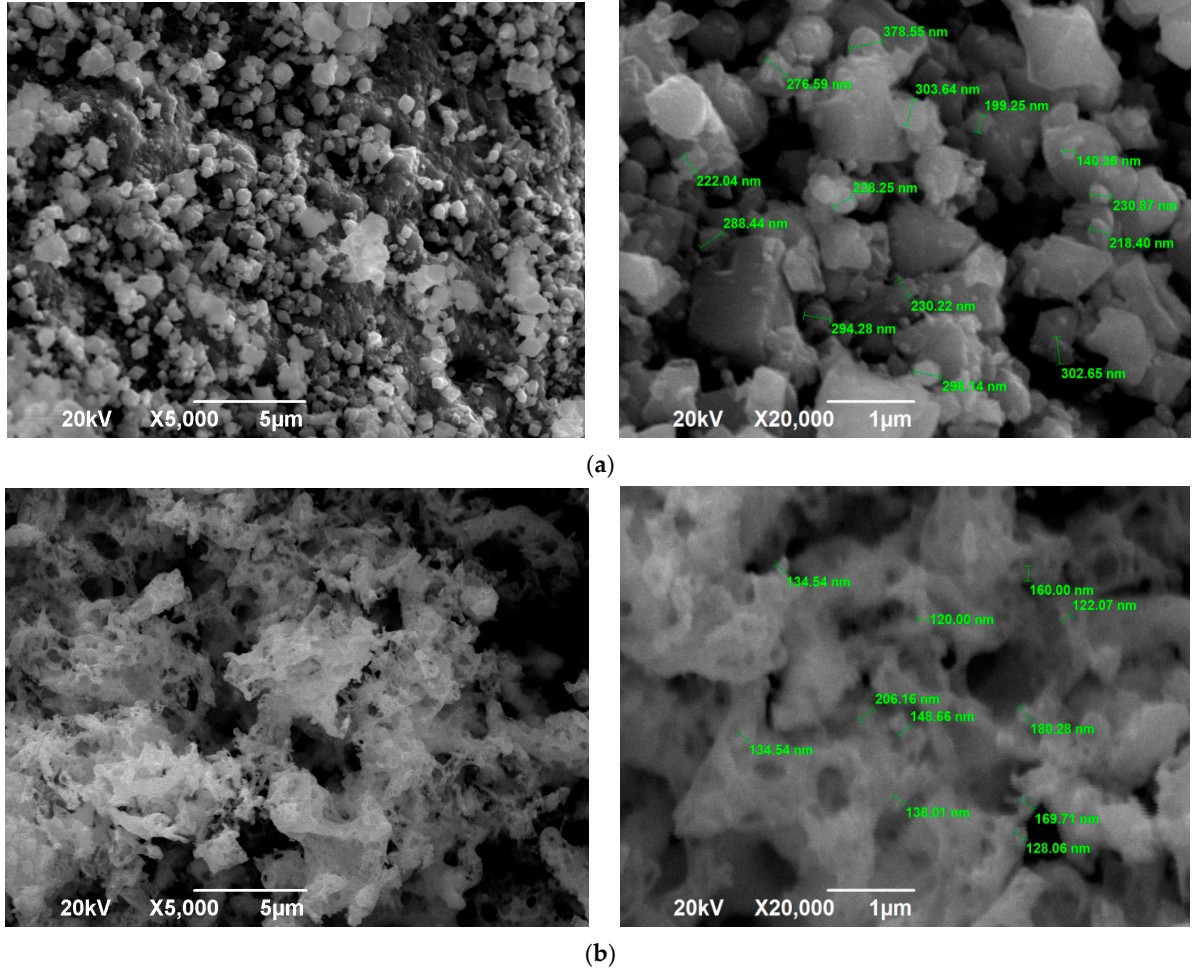

**Figure 3.** Cont.

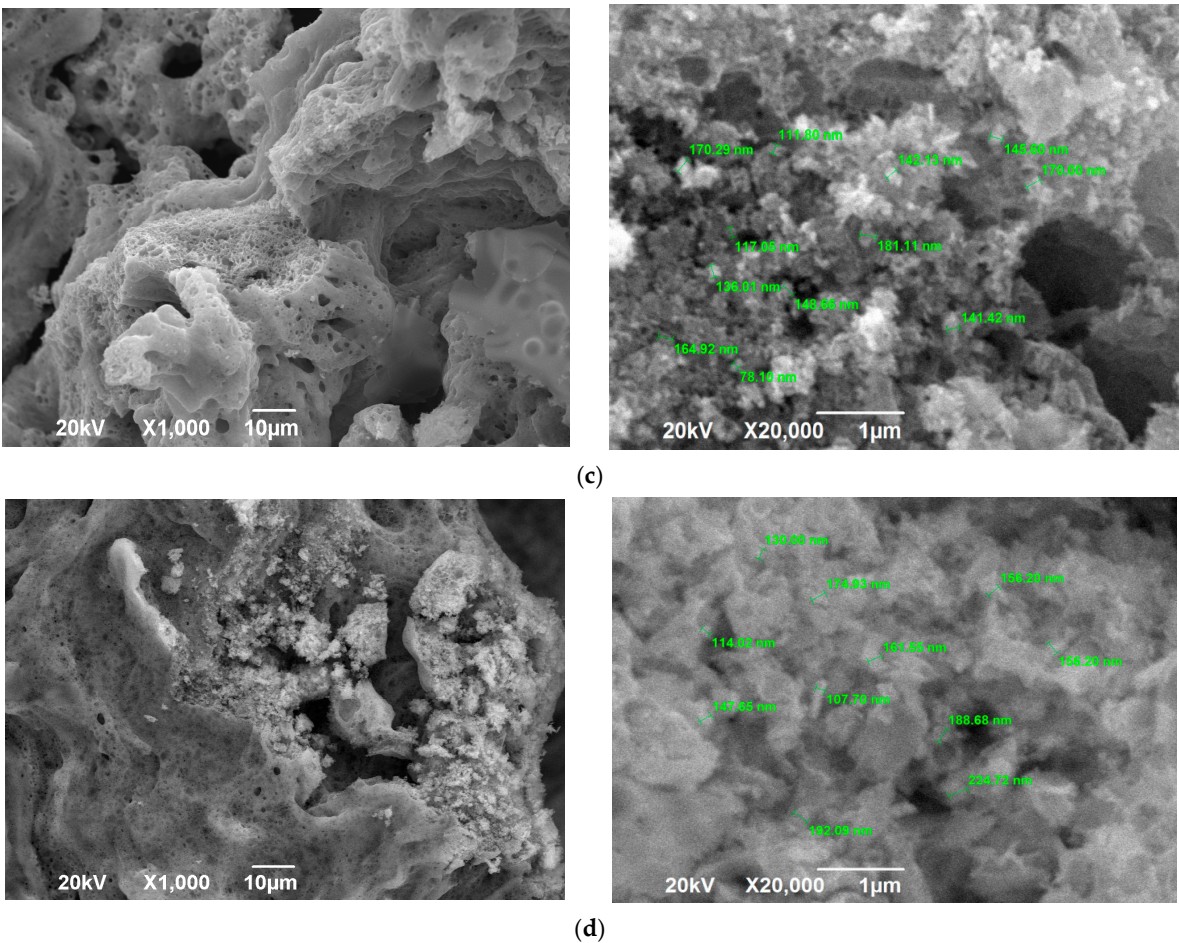

**Figure 3.** SEM images of the SCS product at various φ: (**a**) φ = 0.25; (**b**) φ = 1; (**c**) φ = 2; (**d**) φ = 3.

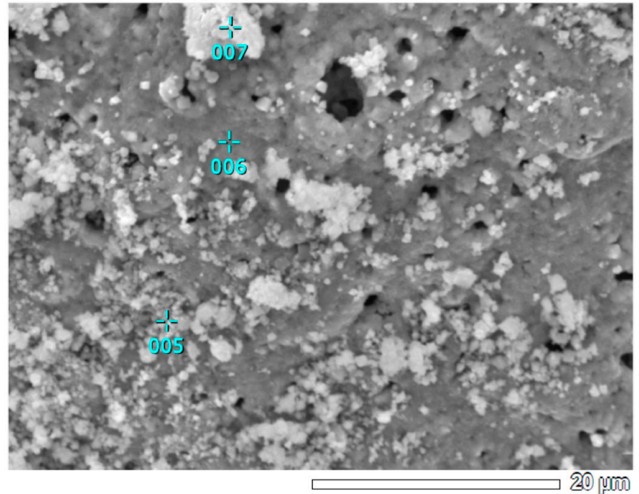

| Element | Element content (wt.%) in the numbered spots | | |
|---------|:---:|:---:|:---:|
| | 005 | 006 | 007 |
| C | 6.07 | 4.92 | 8.36 |
| O | 12.09 | 10.83 | 31.06 |
| Zn | 81.83 | 84.25 | 60.58 |

**Figure 4.** EDS analysis of the local elemental composition of the SCS product at φ = 0.25.

The XRD results for the minimum value φ = 0.25 (Figure 2a) show that the synthesis product consists of two phases: zinc oxide (ZnO) and free carbon (C). To make it easier to evaluate and distinguish the characteristic peaks of ZnO and C, the signs ● and ▼ are added to them, respectively. The presence of free carbon can be explained by the fact that at a minimum value of φ, the combustion temperature is minimal and not high enough for the complete oxidation of glycine even with an excess of oxygen, which leads to the appearance of free carbon in the combustion product [24,27]. High, narrow reflection peaks of ZnO indicate the formation of good crystalline ZnO powders obtained by the synthesis. The sizes of the crystallites determined by calculation according to the Scherrer formula are 22, 26, and 18 nm at the three peaks with the highest intensity, and the average size of the crystallites is 22 nm. It can be seen from Figure 3a that the resulting powder has a homogeneous structure with non-agglomerated, clearly defined particles of equiaxed crystals of submicron sizes. Thus, it can be concluded that an ultrafine powder with a particle size of less than 1 µm, consisting of a mixture of nanoscale and submicron particles, has been synthesized. The results of the EDS analysis of the local elemental composition of this powder at three spots are shown in Figure 4. These results show content from 4.92 to 8.36 wt.% of the carbon in combustion product, which corresponds to the results of the XRD analysis in Figure 2a for the presence of free carbon in the product.

When φ increases from 0.5 up to φ = 3, XRD patterns show the presence of only ZnO; free carbon is no longer detected (Figure 2b–d) due to the formation of free carbon in amorphous form and bound carbon in unburned carbonaceous fragments of organic fuel residue after pyrolysis [20,27]. (Since the XRD patterns in Figure 2b–d show the presence of only the ZnO phase, the ● sign is not added to the characteristic peaks of ZnO.) An increasingly smooth transition from the level of the main background to the level of peaks, which are becoming wider and weaker, indicates the appearance and the increasing share of an amorphous component in the synthesis product. This is confirmed by the structure of the synthesized product in the form of a solidified foam with great many pores of various diameters and agglomerates of small nanoscale and submicron particles that are difficult to distinguish (Figure 3b–d). The amorphous component increased due to the solidification of a gel-like residue with intense gas evolution. An increase in the width of the lower part of the peaks indicates the presence of smaller crystallites with an average size of 13 nm at φ = 1 and 6 nm at φ = 2 in compliance with the Scherrer formula. According to XRD data (Figure 2), a single-phase ZnO with a wurtzite crystal structure of P63mc space group is formed by the SCS reactions at all values of the φ parameter.

The results of the EDS determination of the local carbon content in the SCS product for different values of the molar ratio of fuel to oxidizer (φ) are shown in Figure 5, where the three lines correspond to the local carbon content at three different spots of the combustion product at the same φ.

As can be seen in Figure 5, there is a clear pattern of increasing carbon content in the combustion product from 6% at φ = 0.25 to 30% at φ = 3, which indicates an increase in the content of unburned fragments of organic fuel with an increase in the fuel content in the initial mixture of reagents.

### 3.3. Combustion Products after Calcination

The contamination of the target zinc oxide with high carbon content in combustion products can be eliminated by calcinating (annealing) these products in an oxidizing air atmosphere [20,25,27,30]. The SCS products for φ = 1, 2, and 3 were calcined for 2 h at a temperature of 650 °C, often used in studies of ZnO nanopowders. Upon calcination in air at 650 °C, all samples of the SCS products became white, which indicated the burning away of the free and bound carbon from the zinc oxide. The XRD patterns of the calcined products for φ = 1, 2, and 3 turned out to be almost identical and are shown in Figure 6 for φ = 3.

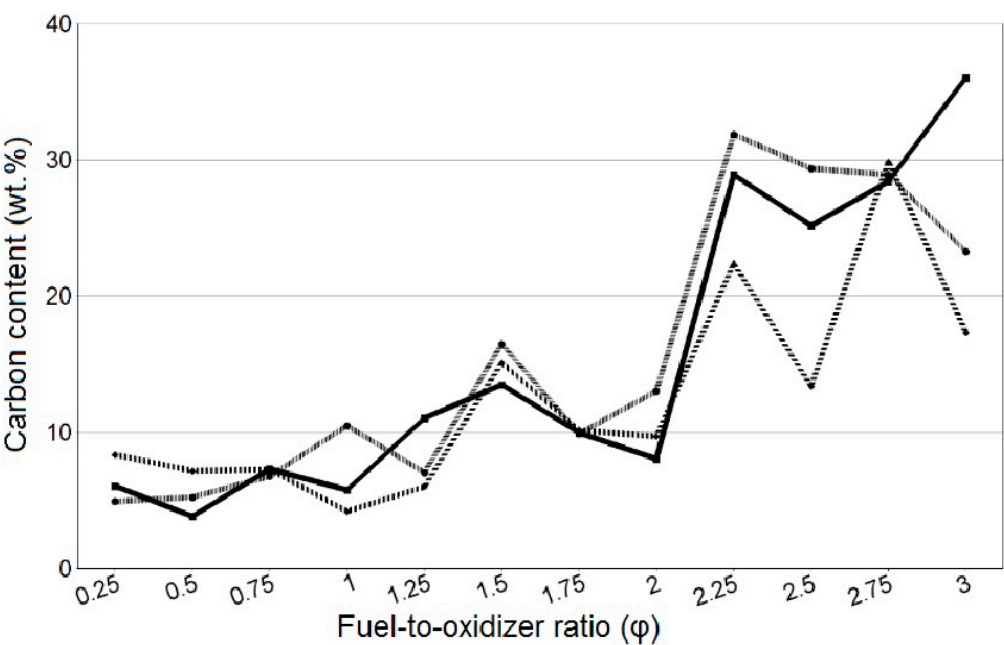

**Figure 5.** Local carbon content (wt.%) at three spots of the SCS product at different $\varphi$.

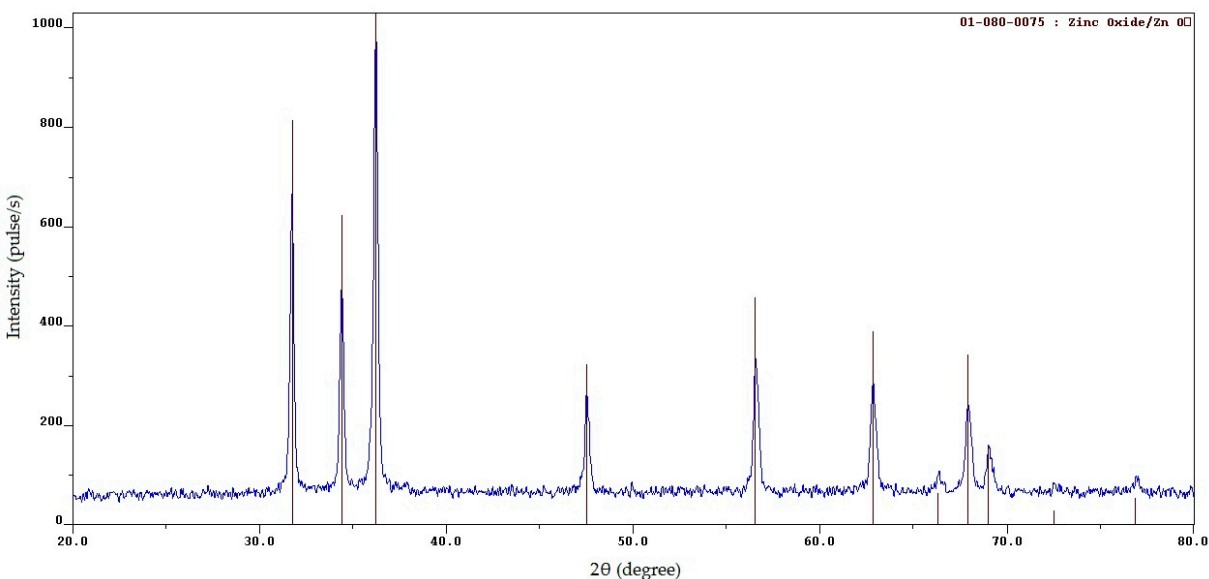

**Figure 6.** XRD patterns of the SCS product at $\varphi = 3$ after calcination.

From the comparison of Figure 6 with Figure 2b–d, it is clearly seen how the XRD patterns of SCS products differ significantly before and after calcination. After calcination, they have high, narrow reflection peaks indicating the presence of a well-formed crystal structure in the resulting zinc oxide with a significantly larger average crystallite size of 44 nm for each $\varphi$. The local carbon content at three spots of the products after calcination was (wt.%): 1.78, 2.52, and 1.77 at $\varphi = 1$; 0.64, 0.55, and 0.81 at $\varphi = 2$; and 1.02, 0.35, and 1.22 at $\varphi = 3$, all respectively. That is, carbon content was on average about 1 wt.%, which is significantly less (especially at large $\varphi$) compared with the carbon content of 5–30 wt.% in the products before calcination (Figure 5). The microstructure of the SCS products after calcination at 650 °C for 2 h is shown in Figure 7.

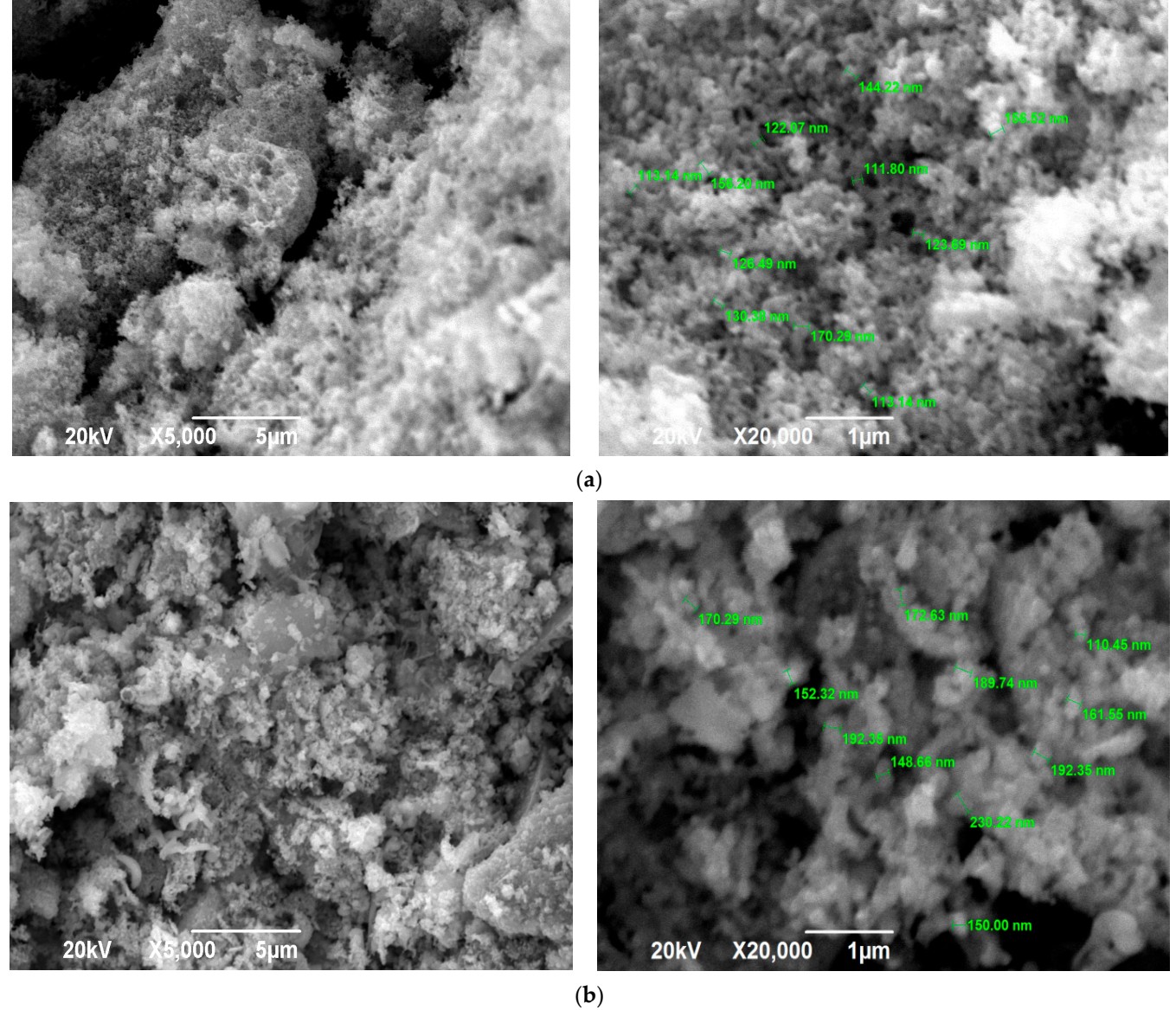

**Figure 7.** SEM images of the SCS product at various φ after calcination: (**a**) φ = 2; (**b**) φ = 3.

The SEM images in Figure 7 for φ = 2 and φ = 3 practically do not differ, which confirms the earlier conclusion about the formation of almost the same product for φ = 1, 2, and 3 after calcination for 2 h at a temperature of 650 °C based on the identity of the product diffractograms for φ = 1, 2, and 3. Figure 7 shows a transition from the structure before calcination (Figure 3) in the form of a solidified amorphous foam with a great many pores of various diameters and agglomerates of small nanoscale and submicron particles that are difficult to distinguish to a more homogeneous structure of a powder-porous body of agglomerates of clearly defined crystalline nanoscale and submicron particles of various shapes: lamellar, equiaxed, rounded.

The research results for the particle-size distribution of the powder SCS product for different φ after calcination by the laser diffraction of aqueous suspensions are shown in Figure 8. The captions indicate the median diameters of $D_{50}$ particles. Particles in aqueous suspensions were subjected to ultrasonic treatment until their size stopped changing.

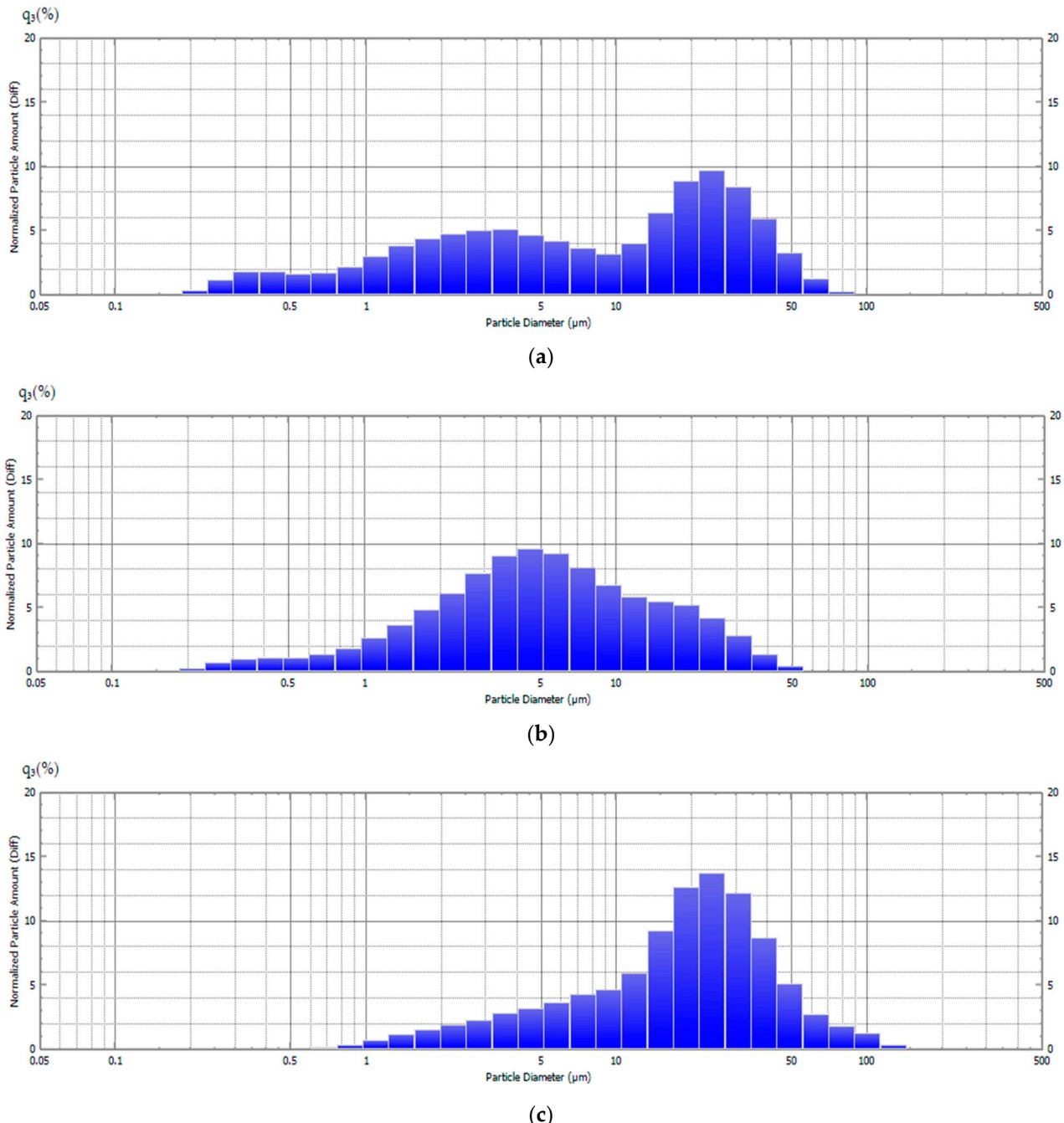

**Figure 8.** Particle-size distribution of the powder SCS product at various $\varphi$ after calcination: (**a**) $\varphi = 0.25$, $D_{50} = 9.0$ μm; (**b**) $\varphi = 1$, $D_{50} = 5.1$ μm; (**c**) $\varphi = 2$, $D_{50} = 19.8$ μm.

Figure 8 shows that these are mixtures of submicron and micron particles from 0.2 to 100 μm, which does not correspond to the sizes of ultrafine (<1 μm) nanoscale and submicron ZnO particles in the images obtained using a scanning electron microscope (Figure 7). This difference can be explained by the fact that the particles in the aqueous suspension are both ultrafine ZnO particles and strong sintered porous agglomerates from 0.2 to 100 μm in size isolated from nanoscale and submicron ZnO particles that are not separated by ultrasonic treatment in water. Thus, after calcination, the SCS product is a powder mixture of individual ultrafine ZnO particles and porous agglomerates up to 100 μm in size sintered from the ultrafine ZnO particles with an average crystallite size of 44 nm.

### 3.4. Band Gap Energy and Photocatalytic Activity of ZnO Powders

The results of measurements of the band gap energy of ZnO powders synthesized by the SCS method at various molar ratios of fuel to oxidizer ($\varphi$) and calcined at various durations at 650 °C are presented in Table 1.

**Table 1.** The band gap energy (eV) of ZnO at various $\varphi$ and calcination duration.

| $\varphi$ | Calcination Duration (h) | | | | | | |
|---|---|---|---|---|---|---|---|
| | 0 | 0.25 | 0.5 | 1 | 2 | 3 | 9 |
| 0.25 | 3.233 | 3.246 | 3.242 | 3.249 | 3.252 | 3.249 | 3.242 |
| 1 | 3.272 | 3.274 | 3.269 | 3.271 | 3.267 | 3.273 | 3.274 |
| 2 | 3.244 | 3.281 | 3.275 | 3.283 | 3.290 | 3.283 | 3.287 |

In general, we can say that the band gap of all powders is identical and ranges from 3.2 to 3.3 eV. Since photocatalysis is carried out under the irradiation of an ultraviolet lamp with a photon energy of 3.37 eV, and the band gap of synthesized ZnO powders is less than this value, all powders must exhibit photocatalytic activity under these conditions. Note that the band gap energy depends more on $\varphi$ than on the duration of calcination. Non-calcined samples with $\varphi = 0.25$ and $\varphi = 2$ showed smaller band gaps than the calcined ones. As a rule, the calcination leads to a slight decrease in the band gap due to the ordering of the structure, but in this case the opposite trend was observed. These differences in the band gap may be associated with changes in morphology, particle size, and surface microstructure, as well as with various defective structures of synthesized ZnO powders, which requires additional research [30,31].

The study of the photocatalytic activity of ZnO powders synthesized by SCS at various values of the $\varphi$ parameter and subjected to the calcination of various durations from 0.5 to 9 h at 650 °C was carried out in the decomposition of phenol dissolved in 100 mL of water at an initial concentration of $C_0 = 1$ mg/L under the action of ultraviolet irradiation with a wavelength of 365 nm. The results of this study are shown in Figure 9 as a dependence of the relative concentration (%) of phenol ($C/C_0$) in an aqueous solution on the ultraviolet irradiation time (h) at different $\varphi$ after different durations of ZnO calcination (the numbers at the lines denote the duration of powder calcination (h), and the number 0 denotes the powder not subjected to calcination).

It can be seen from Figure 9 that the calcination significantly increases the photocatalytic activity of the synthesized ZnO powder, and the longer the calcination, the more significant. In cases where ZnO powder was used without calcination (0) or after a short calcination (0.5), the graphs even show a slight increase in the concentration of phenol by several percent more than 100% in the initial period of the experiment with a decrease in the concentration of phenol as the duration of irradiation increased. This increase in concentration above 100% is associated with the release of unburned organic fuel residues from porous agglomerates of ZnO particles into the solution. These residues have fluorescent properties analogous to phenol, and the total concentration of these residues and phenol is reflected on the graph and could exceed 100%. However, in cases where ZnO powder was used after prolonged calcinations, leading to significant reductions in the remains of unburned organic fuel in the porous agglomerates of ZnO particles, no excess of 100% is observed. The longer the calcination, the less remains of unburned fuel, the better the crystallinity of ZnO powders, and the faster the concentration of phenol decreases during irradiation. At $\varphi = 0.25$ and 1.0, the activity is maximal in ZnO powders after 9 h of calcination. The photocatalytic activity does not depend as much on $\varphi$ as on the duration of calcination, but at $\varphi = 2$, the too-long calcination of the powder slightly reduced its photocatalytic activity; here, the greatest activity was observed in the powder after calcination for durations of 1 and 3 h, which can be related to the enlargement of the ZnO powder during too-long calcination.

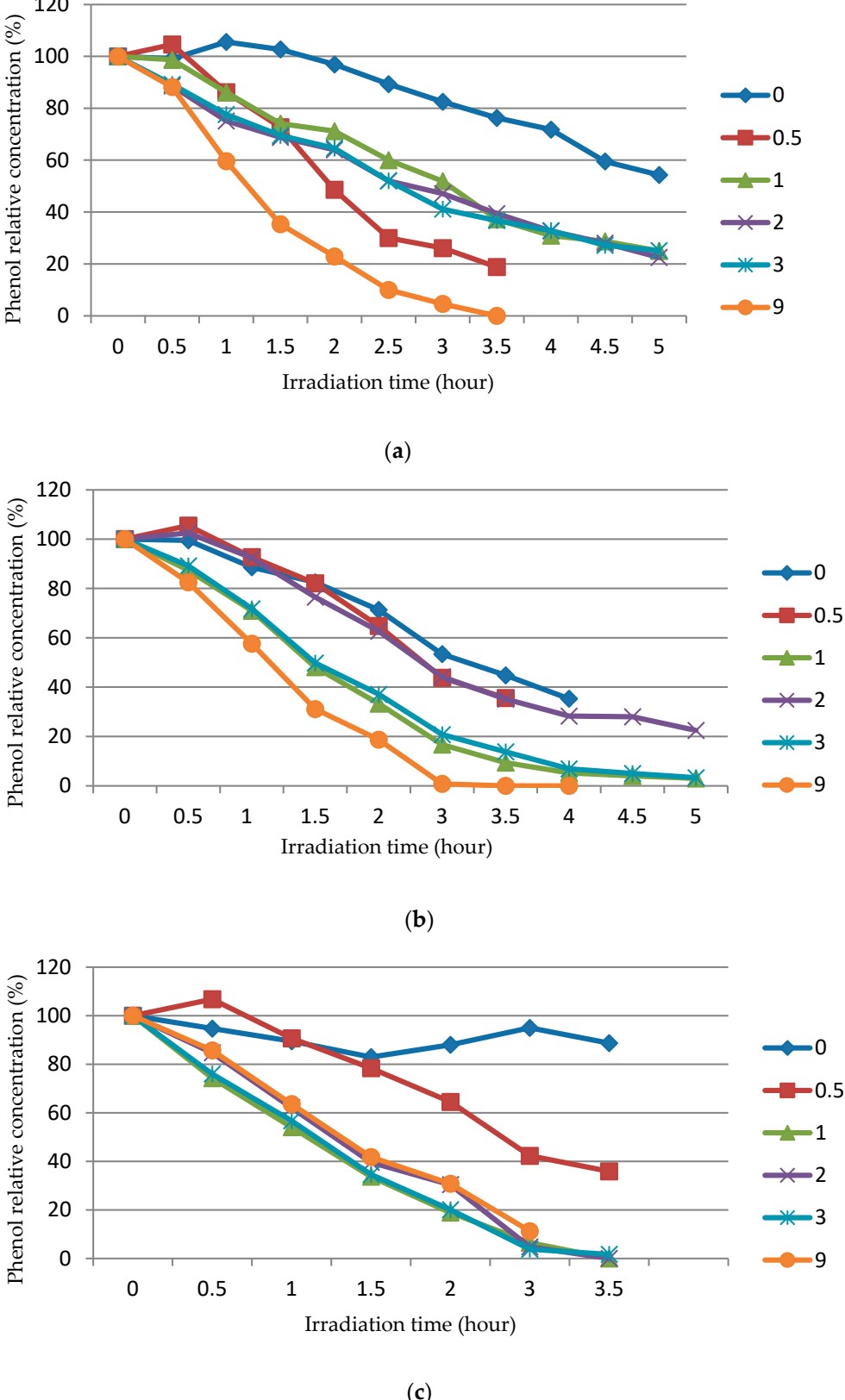

(**a**)

(**b**)

(**c**)

**Figure 9.** The dependence of the relative concentration (%) of phenol ($C/C_0$) in an aqueous solution on the ultraviolet irradiation at different φ after different durations of ZnO calcination (numbers at the line designate hours): (**a**) φ = 0.25; (**b**) φ = 1; (**c**) φ = 2.

In principle, a decrease in the concentration of phenol under the action of ultraviolet irradiation can also occur due to photolysis, that is, a chemical reaction of the decomposition of molecules of a chemical compound under the action of photons. In this regard, the photolysis activity was measured in an experiment. To do this, the phenol solution was subjected to ultraviolet irradiation in the absence of ZnO powder. The results of the experiment are presented in Table 2. As can be seen from Table 2, during the first 4 h of irradiation, the concentration of phenol practically does not decrease, and only at 24 h does it decrease by 14.6%. Thus, photolysis can be ignored when discussing the results of experiments obtained during 5 h of irradiation and presented in Figure 9.

**Table 2.** The dependence of the relative concentration (%) of phenol ($C/C_0$) in an aqueous solution on the ultraviolet irradiation time (h) in the absence of ZnO particles in the solution.

| | The Relative Concentration (%) of Phenol ($C/C_0$) | | | | | |
|---|---|---|---|---|---|---|
| Irridation time (h) | 0 | 1 | 2 | 3 | 4 | 24 |
| $C/C_0$ (%) | 100 | 98.4 | 97.5 | 98.3 | 99.0 | 85.4 |

The possible influence of another factor on the change in the concentration of phenol in the solution was also considered. The adsorption of phenol molecules on the surface of reactor walls and on the surface of ZnO particles is an important phenomenon that is produced at the same time as photodegradation. In order to separate the adsorption effect from the photocatalytic impact, special experiments were conducted in which the phenol solution was in contact with the solid particles under the dark regime for several hours. The results of these experiments are presented in Figure 10.

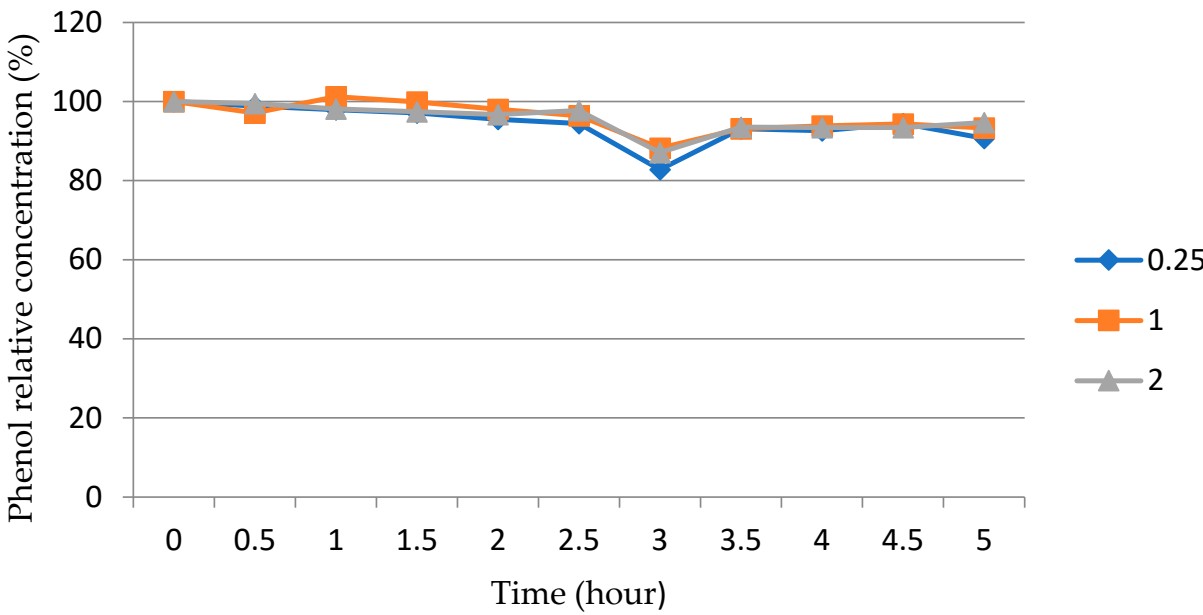

**Figure 10.** The dependence of the relative concentration (%) of phenol ($C/C_0$) on the time (h) under the dark regime in an aqueous solution in contact with ZnO particles sinthesized at different φ (numbers at the line designations) after calcination at 650 °C for 1 h.

Figure 10 shows the same slow decrease in phenol concentration over time in a solution with various synthesized and calcined ZnO powders under the dark regime, that is, in the absence of ultraviolet irradiation, reaching about 10% in 5 h. As follows from Figure 9, under the action of ultraviolet irradiation, the concentration of phenol in a solution with calcined ZnO powders decreases significantly faster and can reach almost zero, that is, indicating the almost complete decomposition of phenol in less than 4 h. During this time, the decrease in the concentration of phenol due to photolysis is 1–2.5% according to Table 2.

Thus, from the comparison of the results on the change in the concentration of phenol in Figures 9 and 10 and in Table 2, it follows that the almost complete decomposition of phenol in less than 4 h occurs predominantly (by $\geq$87.5%) due to the high photocatalytic activity of ZnO but not due to the processes of photolysis and adsorption.

To confirm the photocatalytic degradation of phenol, the presence of by-products of phenol oxidation in the degradation phenol solution after the photocatalytic reaction was investigated by spectrophotometry. The optical absorption of degraded phenol solution in the wave range between 190 and 1000 nm was studied in a quartz cuvette; the thickness of solution layer was 10 cm. The results of spectrophotometric studies did not show any optical absorption bands, which indicated the absence of any phenol oxidation products in the degraded phenol solution.

Thus, Figure 9 demonstrate the high photocatalytic activity of all the ZnO powders synthesized by the SCS method after calcination, since the use of these powders as photocatalysts can lead to the almost complete decomposition of phenol under the action of ultraviolet irradiation in less than 4 h. Such photocatalytic activity is comparable with the activity in the photocatalytic decomposition of phenol by ZnO powders with a nanoscale substructure obtained by the hydrothermal method after calcination at 650 °C for 3–5 h [32] but with significantly higher performance of the SCS method compared with the hydrothermal method and significantly larger photocatalyst particles. The positive sides of the SCS should also include the presence of carbon impurities in its product, which can contribute to an increase in the photocatalytic activity of the synthesized ZnO and, in addition, the increased defectiveness of the synthesized powders due to the very high heating and cooling rates of combustion products, which can also increase their photocatalytic activity [7,27,33–36]. Sufficiently large sintered porous agglomerates up to 100 μm in size from highly dispersed ZnO particles with their high photocatalytic activity can simplify the possibility of their use in membrane photocatalytic installations for deep water purification by the complete photocatalytic destruction of all organic pollutants in the water under the action of ultraviolet irradiation on a photocatalyst suspended in contaminated water, followed by the separation of the photocatalyst after water purification by simple filtration instead of a much more complex and inefficient membrane separation of the suspended photocatalyst from purified water on a ceramic ultrafiltration membrane with a pore size of 100 nm [5,32,37].

## 4. Conclusions

Thus, the burning mode and the composition of burnt products in the preparation of ZnO by the simple, energy-saving, and highly productive SCS method from an aqueous solution of a mixture of reagents (oxidizer–zinc nitrate and fuel–glycine) are governed primarily by the dimensionless parameter φ, equal to the molar ratio of fuel to oxidizer. When a reaction vessel with an aqueous solution of reagents is heated, after water evaporation and gel formation, the reaction takes place at $0.5 \leq \varphi \leq 1.5$ in an extremely fast volumetric combustion mode with a flame and a sharp, almost complete ejection of the reacting mixture and reaction products from the vessel. With the reduced fuel content at $\varphi = 0.25$, the reaction proceeds in a flameless mode with rapid smoke emission and partial product ejection. With increased fuel content ($\varphi > 1.5$), the combustion occurs in a slow smoldering mode with the preservation of a readily destructible combustion product in the vessel. For practical implementation, the most suitable value is $\varphi = 2$, in which the combustion product is a solidified foam-like body of amorphous unburned carbonaceous fragments of organic fuel residue with a carbon content, on average, of 8.5 wt.% and agglomerates of small nanoscale and submicron ZnO particles. After the calcination in an air environment lasting 2 h at 650 °C, the carbon impurity content is reduced, on average, to 0.67 wt.%, and the combustion product acquires a homogeneous structure of a powder body of porous agglomerates up to 100 μm in size, sintered from clearly defined crystalline nanoscale and submicron ZnO particles with an average crystallite size of 44 nm. The ZnO powder exhibits a high photocatalytic activity leading to the almost complete degradation of phenol

in an aqueous solution under the action of ultraviolet irradiation in less than 4 h. The high photocatalytic activity can be explained by such features of the formation of zinc oxide by the SCS process as carbon doping due to the presence of residual carbon impurities and increased defectiveness due to very high heating and cooling rates in combustion. This ZnO powder consisting of sufficiently large porous agglomerates up to 100 μm in size opens the possibility for its simpler application as a suspended photocatalyst in phenol water purification plants under the action of ultraviolet irradiation due to the separation of the photocatalyst after water purification by simple filtration.

**Author Contributions:** The research work was performed and completed through the contributions of all authors. Conceptualization, methodology and validation, A.P.A. and D.L.M.; synthesis and calcination of the samples, E.M.K. and A.A.T.; SEM, EDS and XRD investigations, V.A.N.; band gap and particle-size distribution investigations, I.M.S.; photocatalytic investigations, N.A.K.; writing—original draft preparation, A.P.A., A.A.T., V.A.N., I.M.S. and N.A.K.; writing—review and editing, A.P.A. and D.L.M. All authors analyzed and discussed the data. All authors have read and agreed to the published version of the manuscript.

**Funding:** This research was funded by the Russian Science Foundation, grant number 22-29-00287.

**Institutional Review Board Statement:** Not applicable.

**Informed Consent Statement:** Not applicable.

**Data Availability Statement:** Not applicable.

**Conflicts of Interest:** The authors declare no conflict of interest.

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
