# Peer review of "The Solution Combustion Synthesis of ZnO Powder for the Photodegradation of Phenol"

_ceramics, doi:10.3390/ceramics5040067_

Round 1
Reviewer 1 Report
The current manuscript is well defined and has a detailed description. Each method has been completely discussed i.e., someone who doesn’t have any knowledge or related experiments could understand the pros and cons of this idea versus other related subjects.
After all these beautiful points, some issues come up that seem to result in rendering the manuscript in the best way without any further objection:
1) It is about figure 1.
* It is better to merge all four parts into one whole figure, close to each other, instead of four separate schemes that have been considered as one figure.
** Part (b) and part (c), both describe the ignition time but in different amounts of molar ratios. However, they have different time units i.e. the (b) is in form of seconds and the (c) is in the form of minutes. It seems to be not apparent for the reader to understand it well.
2) It is about figures 2 to 4.
* It couldn’t be easy to understand the main peaks in the XRD patterns, especially in figure 2. It is difficult to estimate and distinguish the ZnO and C characteristic peaks.
3) As mentioned on page number 6 and 11, by increasing the molar ratios that increase the amount of C content i.e. pure or in bond with O, the system tends to become black. On the contrary, increasing the amount of Zn content which is the reason for lowering the molar ratio number, makes the system white. However, according to figure 2 and figure 4, something reverse or opposite happens.
4) It is about figure 7. What is the difference between phi with the amount of 2 vs 3? (It couldn’t be clearly defined from the SEM images.
Reviewer 2 Report
1- This article presents the photocatlytique activity of ZnO particles obtained by solution combustion synthesis (SCS) of oxides method. In line 109, the authors confirm that SCS is a known technique as per references [20,24-28]. The originality is the application of the obtained ZnO in the photocatalytic degradation of phenol.
2- The authors specify that the recombination of charges, largely due to the presence of impurities, and the large band gap of ZnO slower its photocatalytic activity. On the one hand, their experimental results show that around 5% to 8%wt Fig 4, (local concentration 7% to more than 30%wt) of carbon remains inside with ZnO. This high carbon concentration can constitute recombination centers demonstrating that SCS technique didn’t allow the elaboration of the pure ZnO phase necessary to achieve high-yield pollutant abatement.
3- On the other hand the values of band gap presented in Table 1. range from 3.2 to 3.3 indicating no lowering effect of the band gap due to the presence of carbon(for ZnO band gap value see http://dx.doi.org/10.1063/1.367375). So an improvement in photocatalytic activity will not be expected.
4- More, in relation to the band gap, the authors should explain the experimental procedure they used to measure the band gap of the ZnO powders.
5- The degradation of phenol by these materials obtained under various calcination times and oxidant/combustible ratios, presented in Fig. 9 show severe discrepancies :
a. Several values of relative concentration overrun 100% meaning the production of phenol rather than its destruction without any explanation.
b. No photolyse activity has been measured during the experiments.
c. The adsorption of pollutant molecules on the surface of reactor walls and on the surface of catalysts is an important phenomenon that is produced at the same time as photodegradation. In order to separate the adsorption effect from the photocatalytic impact, the solution should be in contact with the solid particles under the dark regime for several hours utile the stabilization of phenol concentration. Such precautions have not been taken during these experiments so it is not possible to attribute the phenol degradation to the photocatalytic activity of ZnO or to adsorption processes.
d. Measurement of the total carbon balance in the degradation phenol solution should be done to confirm the photocatalytic and photolyse degradation of phenol.
6- Authors write in line 35: “as well as a photocatalyst for oxidation of toxic organic substances to harmless CO2 and H2O [4,5]. This is not exact as some organic pollutants molecules contain N and S elements. So the oxidation will produce some sulfates and nitrates etc.
7- In lines 46 to 49 “electrons are in free (can move through the crystal lattice) and bound (participate in chemical bonds with the ions of the crystal lattice) states. The transfer of an electron from a bound to a free state is associated with energy consumption governed by the band gap energy (3.37 eV for ZnO),” It is usual to use the term “transition of electrons from the valence band to the conduction band…”
8- There are several affirmative sentences without any reference.

Reviewer 3 Report
Dear Authors,
the results of your work are interesting and seems to be relatively high-quality and clearly processed. Nevertheless, the interpretation of the data would be helped by answering following comments:
1) Did you evaluate the carbon content also in samples annealed at different temperatures than 2 hours? How do you explain that the photocatalytic activity maximum is for each φ at a different temperature?
2) How is it possible that ratio C/C0 (% of phenol) is higher than 100 % in some cases? Is it due to measurement error or deviation?
3) Did you also tested UV degradation of phenol without the ZnO powder?
I also recommend minor language corrections, for example:
calcining vs. calcination (better)
duration x irradiation time (better)
Round 2
Reviewer 2 Report
At least the authors show that about 15% of phenol degradation is due to photolysis and about 10% due to adsorption, that is 25% of degradation is not due to photocatalytic activity of ZnO.
More, if they don't see any by-product signature in the post-treatment solution by the spectrophotometric method it doesn't mean that by-products don't exist.
So it is not exact to say that the complete degradation of phenol is due to the photocatalytic activity of ZnO.
